# In Search for Low-Molecular-Weight Ligands of Human Serum Albumin That Affect Its Affinity for Monomeric Amyloid β Peptide

**DOI:** 10.3390/ijms25094975

**Published:** 2024-05-02

**Authors:** Evgenia I. Deryusheva, Marina P. Shevelyova, Victoria A. Rastrygina, Ekaterina L. Nemashkalova, Alisa A. Vologzhannikova, Andrey V. Machulin, Alija A. Nazipova, Maria E. Permyakova, Sergei E. Permyakov, Ekaterina A. Litus

**Affiliations:** 1Institute for Biological Instrumentation, Pushchino Scientific Center for Biological Research of the Russian Academy of Sciences, Institutskaya Str., 7, Pushchino 142290, Moscow Region, Russia; marina.shevelyova@gmail.com (M.P.S.); certusfides@gmail.com (V.A.R.); elnemashkalova@gmail.com (E.L.N.); lisiks.av@gmail.com (A.A.V.); alija-alex@rambler.ru (A.A.N.); mperm1977@gmail.com (M.E.P.); permyakov.se@ya.ru (S.E.P.); ealitus@gmail.com (E.A.L.); 2Skryabin Institute of Biochemistry and Physiology of Microorganisms, Pushchino Scientific Center for Biological Research of the Russian Academy of Sciences, Pr. Nauki, 5, Pushchino 142290, Moscow Region, Russia; and.machul@gmail.com

**Keywords:** Alzheimer’s disease, amyloid β peptide, human serum albumin, low-molecular-weight ligand, protein–ligand interaction, surface plasmon resonance spectroscopy

## Abstract

An imbalance between production and excretion of amyloid β peptide (Aβ) in the brain tissues of Alzheimer’s disease (AD) patients leads to Aβ accumulation and the formation of noxious Aβ oligomers/plaques. A promising approach to AD prevention is the reduction of free Aβ levels by directed enhancement of Aβ binding to its natural depot, human serum albumin (HSA). We previously demonstrated the ability of specific low-molecular-weight ligands (LMWLs) in HSA to improve its affinity for Aβ. Here we develop this approach through a bioinformatic search for the clinically approved AD-related LMWLs in HSA, followed by classification of the candidates according to the predicted location of their binding sites on the HSA surface, ranking of the candidates, and selective experimental validation of their impact on HSA affinity for Aβ. The top 100 candidate LMWLs were classified into five clusters. The specific representatives of the different clusters exhibit dramatically different behavior, with 3- to 13-fold changes in equilibrium dissociation constants for the HSA–Aβ40 interaction: prednisone favors HSA–Aβ interaction, mefenamic acid shows the opposite effect, and levothyroxine exhibits bidirectional effects. Overall, the LMWLs in HSA chosen here provide a basis for drug repurposing for AD prevention, and for the search of medications promoting AD progression.

## 1. Introduction

Human serum albumin (HSA) is the predominant protein in blood plasma, accounting for approximately 60% of the total protein content [1]. HSA is a 66 kDa protein, containing 585 amino acid residues organized into three domains [2]. It possesses seven fatty acid (FA) binding sites and the two major sites specific to small molecules [3,4]. HSA exhibits a remarkable ability to bind and transport through the bloodstream numerous endogenous and exogenous ligands [5,6]. Thereby HSA serves as a depot and delivery vehicle for small molecules in the blood, such as FAs, hormones, bilirubin, hemin, drugs, etc. [6,7,8,9]. The long plasma half-life of HSA (12.7–18.2 days [2]) is widely used to prolong the half-life of therapeutic peptides/proteins by their covalent modification with FAs or other substances with high affinity for HSA [10]. Some of them are used for the treatment of diabetes mellitus (type 1 and 2) and/or obesity [11,12]: insulin detemir (Levemir^®^), insulin degludec (Tresiba^®^), liraglutide (Victoza^®^/Saxenda^®^), and semaglutide (Ozempic^®^, Rybelsus^®^, Wegovy^®^). Note that the binding of various ligands by HSA is actively studied by various in silico approaches [13,14,15].

HSA is a natural depot for amyloid β peptide (Aβ) [16,17], one of the key factors in the development of Alzheimer’s disease (AD) [18,19]. HSA binds about 90% of Aβ in blood serum [20] and, according to various estimates, from 40% to 94% in cerebrospinal fluid [21]. Moreover, it is speculated that targeting HSA in the peripheral bloodstream may shift the balance towards removal of Aβ from the central nervous system [22,23]. HSA inhibits Aβ aggregation and lowers the risk of AD and its progression [22,24,25,26,27]. As a component of the interstitial fluid, HSA is presented in the intercellular space of the brain parenchyma [28]. Although HSA is primarily synthesized in the liver, brain microglial cells also synthesize and secrete HSA, especially upon their stimulation with Aβ or lipopolysaccharide [29]. Finally, HSA is included in amyloid deposits (plaques) in the brain of AD patients [30]. These facts indicate the significance of HSA in Aβ metabolism and the potential of HSA usage for therapy of AD.

Preliminary clinical trials confirm the effectiveness of AD treatment by replacement of the patient’s serum albumin with its pharmacological preparation via plasmapheresis (plasma exchange (PE)) [22,27]. This approach is related to a risk of cardiovascular and respiratory complications, anaphylactoid reactions, infections, and hemorrhage [31]. Among the adverse events encountered in AD patients and associated with PE are hypotension, muscle spasms, syncope, anemia, device-related infections, and some others [27,32]. The necessity to live with a catheter inserted in the chest and metabolic alterations related to PE can trigger psychiatric symptoms in the PE-treated AD patients [27]. Nevertheless, this approach is considered relatively safe, but requires the participation of highly trained medical personnel [27,33,34].

The potentially less harmful approach is to increase HSA affinity for Aβ via allosteric action of endogenous or exogenous HSA ligands, initially demonstrated for linoleic and arachidonic acids [35]. We further showed the even more pronounced effects for serotonin [36] and ibuprofen [37]. Moreover, ibuprofen enhances HSA ability to inhibit Aβ fibrillation [37]. These data are in line with clinical observations and the results derived from animal models of AD [38,39,40].

Despite the first encouraging results, a systematic search of the HSA ligands that can affect its interaction with Aβ has not yet been carried out. To fill this gap, in the present work, we systematically search for the clinically approved low-molecular-weight ligands (LMWLs) in HSA related to AD progression that could modify HSA affinity for Aβ, and selectively test the candidate ligands from this perspective after their careful structural systematization and ranking. We have focused primarily on the low-molecular-weight drugs (or “small molecules”), since they are highly bioavailable and cross the blood–brain barrier (BBB) relatively easily, which is critical for their action in the intercellular space of the brain parenchyma.

## 2. Results

### 2.1. Bioinformatic Selection of the Therapeutic LMWLs Associated with AD

The DrugBank database [41,42] was used as a source of the clinically approved HSA ligands related to AD. For each of the HSA ligands we have collected a set of the characteristics necessary for their further filtering: molecular mass values, experimental and theoretical water solubility, and BBB penetration. Information on plasma concentrations of the substances was manually collected from Pubmed and DrugBank. The association of the substances with AD was assessed by the number of the relevant literature sources found on the Alzforum online resource. The filtering of the candidates by a set of parameters: molecular mass above 100 Da and less than 900 Da, water solubility above 1 μM, predicted BBB penetration exceeding 50%, plasma concentration above 0.5 nM, and more than 1 reference on Alzforum, resulted in 100 LMWLs in HSA associated with AD (Appendix A), after removal of the FAs studied in our previous works [9,15,35,43]. The candidate compounds belong to different drug classes, including such common ones as antidiabetic drugs (rosiglitazone), non-steroidal anti-inflammatory drugs (ibuprofen, meloxicam), neuroleptics (risperidone), vitamins (vitamin A, thiamine), antibiotics (tetracycline, ampicillin), and hormones (testosterone, estradiol).

### 2.2. Classification and Ranking of the Selected LMWLs According to the Expected Location of their Binding Sites on HSA

To predict the location of the binding sites for the 100 AD-related LMWLs on the HSA surface, three-dimensional structures of the HSA–LMWL complexes were built using Auto Dock Vina [44]. Note that the binding sites were previously experimentally localized by X-ray only for a few members of our panel, namely ibuprofen [3], warfarin [3], indomethacin [3], halothane [45], and propofol [45]. The binding sites predicted for these ligands coincide with those localized experimentally for all ligands, except for ibuprofen (the predicted site is located near the secondary binding site). The resulting binding sites were hierarchically classified into five clusters based on the dynamic time warping algorithms as implemented in the R language library ‘dtw’, used for classification and clustering of general number series (Appendix A) [46]. The division of the binding sites into a larger number of groups led to fragmentation of the already identified five groups into smaller ones with very similar characteristics but did not lead to a more detailed localization of the clusters and was therefore considered inappropriate.

Figure 1 illustrates the location of the key residues constituting these clusters on HSA molecule. Cluster I is located between HSA subdomains IIA and IIB and is predicted to be specific for the 34 AD-associated LMWLs. The clusters II (subdomain IIIA), III (subdomain IB), and V (subdomain IIA) are predicted to bind 24, 13, and 28 ligands, respectively. Levothyroxine (DB00451) was attributed to a distinct cluster IV located between the cleft and subdomain IB.

The amino acid residues most commonly involved in recognition of the ligands (the residues with highest *f_aa_* values, see Equation (1), Figure 1):Cluster I: P447 (*f_aa_* = 84.4%), R222 (78.1%), K444 (62.5%), D451 (59.4%), and E292 (46.9%);Cluster II: Y452 (*f_aa_* = 40.5%), K436 (32.4%), K195 (32.4%), A191 (27.0%), and K432 (27.0%);Cluster III: R117 (*f_aa_* = 39.0%), E141 (34.8%), R145 (30.4%), E86, P118 (26%), and Y140 (26%);Cluster IV: R145, E425, N109, Q459, H146, R114, K525, and R186;Cluster V: E292 (*f_aa_* = 55.6%), R257 (47.2%), K199 (44.4%), R218 (41.7%), and L219 (38.9%).

To select the most representative ligands in each cluster, the ranks of the ligands within each cluster (*R*) were calculated according to Equation (1), followed by choice of the ligands with high *R* values (Figure 1). The resulting ligands are most often recognized by the aforementioned residues, thereby serving as the most representative candidates for further experimental examination of their influence on HSA–Aβ interaction. Among them, only one candidate from each cluster was taken for further consideration, with an emphasis on the well-studied, socially significant drugs (Table 1). Most of the selected candidates (prednisone, warfarin, levothyroxine, and propranolol) are among the top 100 most commonly prescribed drugs in the USA (according to the ClinCalc DrugStats Database, https://clincalc.com/DrugStats/Top300Drugs.aspx; accessed on 14 March 2024). The number of the AD-associated literature sources for the selected candidates ranges from five to twenty-four (for R-warfarin, the number of the references reaches 139717; for warfarin—18). The maximum peak blood concentration (C_max_) of the candidates ranges from 0.1 µg/mL to 0.91 µg/mL. Solubility of the candidates (except for levothyroxine) in the analysis buffer exceeded 200 µM.

Glucocorticoid prednisone is a four-ring molecule containing five oxygen atoms in keto-, hydroxyl- and carboxy-groups ((8S,9S,10R,13S,14S,17R)-17-hydroxy-17-(2-hydroxyacetyl)-10,13-dimethyl-6,7,8,9,12,14,15,16-octahydrocyclopenta[a]phenanthrene-3,11-dione). The main predicted binding site (LYS444, PRO447, TYR452, ARG218, ARG222, and ASP451) includes three cationic, one hydrophobic, one aromatic, and one anionic amino acid residues (for more details see Appendix A).

Warfarin (2-hydroxy-3-(3-oxo-1-phenylbutyl)chromen-4-one) is a cyclic ester 4-hydroxycoumarin substituted by a 1-phenyl-3-oxo-1-butyl group with one chiral center. The molecule contains three rings and four oxygen atoms, two of which are involved in keto-enol tautomerism. The predicted binding site (ASP187, ALA194, LYS195, LYS432, LYS436, VAL455, and TYR452) involves three cationic, one anionic, two hydrophobic, and one aromatic amino acid residues (for more details see Appendix A).

Mefenamic acid (2-[(2,3-dimethylphenyl)amino]benzoic acid) is an aminobenzoic acid-containing dimethylphenyl group. The molecule includes two benzoic rings, one secondary amine group, and one ester group. The binding site (ARG117, PRO118, TYR140, GLU141, LEU182, ARG145, and ARG186) consists of three cationic, one anionic, two hydrophobic, and one aromatic amino acid residues (for more details see Appendix A).

Levothyroxine ((2S)-2-amino-3-[4-(4-hydroxy-3,5-diiodophenoxy)-3,5-diiodophenyl]propanoic acid) is the L-enantiomer of thyroxine, consists of two linked diiodo-substituted phenolic residues and one α-amino group. The predicted binding site (ARG145, GLU425, ASN109, GLN459, HIS146, ARG114, LYS525, and ARG186) contains five cationic, one anionic, and two hydrophilic but non-ionic amino acid residues (for more details see Appendix A).

Propranolol (1-(naphthalen-1-yloxy)-3-[(propan-2-yl)amino]propan-2-ol) contains a naphthalene group, two oxygen atoms in ether and hydroxyl groups, and one nitrogen atom in a secondary amine group. The propranolol molecule contains one chiral center. The binding site (LYS195, LYS199, TRP214, ALA215, ARG218, LEU238, ALA291, GLU292, SER192, GLN196, HIS242, and ARG257) contains five cationic, three hydrophobic, one aromatic, one anionic, and two hydrophilic, but non-ionic amino acid residues (for more details see Appendix A).

### 2.3. Experimental Validation of the Ability of the Top-Ranked AD-Related LMWLs to Affect HSA Affinity for Aβ40

The effect on HSA interaction with monomeric Aβ40 in the following selected representatives of the clusters I-V was studied using surface plasmon resonance (SPR) spectroscopy: prednisone (cluster I), warfarin (cluster II), mefenamic acid (cluster III), levothyroxine (cluster IV), and propranolol (cluster V). The ligand concentrations were chosen to ensure their efficient binding to HSA, based on the literature data on HSA affinity for the ligands (Table 1). With the exception of levothyroxine, the estimated occupancy of HSA binding sites exceeded 70%. Due to the lower solubility of levothyroxine in the assay buffers (15 μM), the achievable occupancy of HSA binding sites ranged from 9% to 41%. The resulting SPR sensograms for 20/40 μM HSA and their description using the heterogeneous ligand model (Equation (2)) are shown in Figure 2. The respective averaged kinetic and equilibrium dissociation/association constants are represented in Table 2. The most pronounced ligand binding-induced changes in HSA affinity for monomeric Aβ40 in the presence of ethanol are observed for mefenamic acid with an increase in the *K_D_* values by a factor of 3–4. This effect is clearly seen in the scale of free energy changes accompanying the HSA–Aβ40 interaction (Figure 3). Similarly, the most marked changes in the presence of dimethyl sulfoxide (DMSO) are observed for levothyroxine (3-fold increase in the *K_D_*_1_ value and 12-fold decrease in the *K_D_*_2_) and prednisone (decrease in the *K_D_*_2_ value by a factor of 13) (see Figure 3, Table 2).

Taken together, the SPR data demonstrate the ability of some representative members of the specific clusters (namely I, III, and IV) to affect HSA affinity for Aβ40. Meanwhile, one should expect various ligand- and cluster-specific effects, as exemplified by ibuprofen (cluster II [37]) and serotonin (cluster V [36]).

## 3. Discussion

The approved medications are a valuable source of the substances potentially suited for therapy of both common and rare diseases within the approach known as drug repurposing or repositioning [53]. For example, tetracyclines and some polyphenols are able to interfere with aggregation of several amyloidogenic proteins, such as amyloid polypeptide (type-2 diabetes) and transthyretin (senile systemic amyloidosis, familial amyloid polyneuropathy, and cardiomyopathy) [54,55,56]. Similarly, some of the approved drugs have the potential to worsen certain disorders. In the present work, we systematically searched for the approved medications able to affect specificity to Aβ in HSA as a major natural depot for Aβ in the blood/CSF [21], with a focus on the HSA ligands associated with AD. In our previous works we identified several such HSA ligands [35,36,37], but here we used a general bioinformatic approach that allowed us to obtain a panel of 100 drug candidates (Appendix A). They were grouped into five clusters according to the predicted location of their binding sites on HSA molecule, followed by ranking of the candidates within each cluster (Appendix A). The massive involvement of polar and charged amino acid residues in the predicted binding sites of the ligands on HSA (Appendix A) indicates potential dependence of the ligand–HSA interactions on pH and ionic strength of a solution. Meanwhile, it should be kept in mind that only one model of a ligand–HSA complex with the lowest energy was taken into account, while ligand binding to HSA at several distinct sites was shown for a number of substances (ibuprofen, halothane, propofol, and others).

The developed panel of 100 LMWLs in HSA represents a valuable source of the approved drugs with the potential to affect AD progression due to their ability to modulate HSA–Aβ interaction. To probe this suggestion, one representative member from each cluster was selected (Table 1) for experimental validation of their ability to affect Aβ40 binding to HSA, focusing on their outstanding clinical value. The clinical data (Table 3) [57,58,59,60,61,62,63], as well as the results derived from animal models and in vitro studies [64,65,66], evidence an association of the selected drugs with AD. Meanwhile, the influence of the candidates on Aβ metabolism and HSA interaction with Aβ has not been reported to date.

The clusters III and IV overlap with the Aβ binding site predicted for HSA [37] and confirmed in ref. [67] (the region between domains IB and IIIB), while the remaining clusters are more or less close to it. Therefore, the ligands belonging to these clusters may directly, allosterically, or both, affect the HSA–Aβ40 interaction. This suggestion is supported by the data presented in Table 3: several members of the different clusters exert different effects on HSA affinity to Aβ. For instance, mefenamic acid (cluster III) favors dissociation of HSA-Aβ40 complex, while prednisone (cluster I) promotes this interaction. Furthermore, the effects for ligands within the same cluster may be opposite, as exemplified by prednisone and risperidone. At the same time, according to the previous data [36,37], we cannot exclude the possibility of direct interaction of HSA ligands with Aβ.

We have previously shown that ibuprofen [37] and serotonin [36] favor HSA interaction with Aβ (Table 3). The analogous effect is shown here for prednisone (Table 2), indicating its potential with regard to prevention of amyloid deposition in the brain. Meanwhile, in a randomized, placebo-controlled multicenter trial low-dose prednisone did not show behavioral improvements compared with the placebo group [68]. The lack of positive clinical data for prednisone may be due to its rapid clearance in the liver, and the problem could be solved by directed improvement of its pharmacokinetic properties. At the same time, the use of high-dose intrathecal corticosteroids has been proposed as a promising approach to AD prevention [58]. In addition, similarly to ibuprofen, prednisone has an anti-inflammatory effect, which is beneficial for prevention of development of neuroinflammation, another pathogenetic factor in AD [69].

On the contrary, the substances that prevent HSA interaction with Aβ, such as mefenamic acid and risperidone (Table 3), should be considered as potentially harmful with regard to stimulation of amyloid deposition in the brain and AD progression. Apparently, in vitro and clinical studies are needed to establish the relevance for AD in these cases. In any case, identification of the drugs that affect HSA–Aβ interaction is important for further studies of their impact on AD, regardless of the direction of the effect.
ijms-25-04975-t003_Table 3Table 3Summary of the effect of the AD-related LMWLs in HSA on its interaction with Aβ and data on the role of these ligands in AD progression.ClusterHSA LigandEffect of the Ligand on HSA Affinity for AβRelevance for AD ProgressionIprednisone↑ ^†^decline of AD biomarkers in non-AD patients after taking prednisone [57,58]; lack of effect in the treatment of AD patients [68]risperidone ↓ [70]reduces psychosis and favors functioning in elderly patients with psychosis of AD and mixed dementia [71]IIibuprofen ↑ [37]reduces the risk of AD progression [39]IIImefenamic acid↓ ^†^not availableIVlevothyroxinebidirectional effect ^†^hypothyroidism, thyroiditis and hyperthyroidism are more common among AD patients [72]; lowered thyroxine level in cerebrospinal fluid of AD patients [61]Vserotonin ↑ [36]modulates Aβ level in the central nervous system of AD patients [40]^†^ See Table 2; ↑—promote interaction; ↓—prevent interaction.


## 4. Materials and Methods

### 4.1. Materials

Recombinant human Aβ40 was expressed in *E. coli* and purified as previously described [37]. Briefly, the supernatant after cell disintegration was loaded onto a Profinity^TM^ IMAC resin column and the 6–His–ubiquitin–Aβ40 fusion protein was eluted with a linear gradient of imidazole. Aβ40 was excised from the fusion protein via treatment by catalytic core of ubiquitin carboxyl-terminal hydrolase 2 (Usp2-cc). To remove the His-tagged ubiquitin and Usp2-cc, the hydrolysate was loaded onto a Profinity^TM^ IMAC resin column and the unbound fraction containing Aβ40 was purified by high-performance liquid chromatography using a Phenomenex^®^ Jupiter C18 column. Precise chain cleavage by Usp2-cc was confirmed by electrospray ionization mass spectrometry (Shimadzu LCMS-2010EV).

Usp2-cc was prepared mainly as described in ref. [73]. The FA-free HSA prepared under non-denaturing conditions [74] was from Merck (cat. #126654, Darmstadt, Germany). Protein concentrations were measured spectrophotometrically at pH 7.4–8.0 and calculated using molar extinction coefficients at 280 nm estimated according to ref. [75]: 34,445 M^−1^cm^−1^ for HSA, 41,370 M^−1^cm^−1^ for Usp2-cc, and 1490 M^−1^cm^−1^ for Aβ40.

Propranolol (cat. #P913470), levothyroxine (L-thyroxine) (J62606.03), prednisone (PHR1042), warfarin (A2250), and mefenamic acid (M4267) were at least 98% pure and purchased from Macklin, Thermo Fisher Scientific (Waltham, MA, USA), and Sigma–Aldrich (Burlington, MA, USA) (for the last three compounds), respectively. Ultra-grade Tris and 2-mercaptoethanol (2-ME) were purchased from Amresco^®^ LLC (Vienna, Austria). Urea, imidazole, sodium chloride, sodium hydroxide, sodium dodecyl sulfate (SDS), DL-dithiothreitol (DTT), and glycerol were from Panreac AppliChem (Darmstadt, Germany). Ethylenediaminetetraacetic acid (EDTA), acetonitrile, N-Ethyl-N′-(3-dimethylaminopropyl) carbodiimide hydrochloride (EDAC), N-hydroxysulfosuccinimide sodium salt (sulfo-NHS), and polyethylene glycol sorbitan monolaurate (TWEEN^®^) 20 were from Sigma–Aldrich (St. Louis, MO, USA). Calcium chloride was from Fluka (Buchs, Switzerland). Ethanolamine and Profinity^TM^ IMAC resin were bought from Bio-Rad Laboratories (Hercules, CA, USA) Hydrochloric acid was from Sigma Tec LLC (Moscow, Russia). DMSO was from Helicon (Moscow, Russia). Trifluoroacetic acid (TFA) was purchased from Fisher Scientific (Pittsburgh, Pennsylvania, U.S.). Potassium chloride and sodium azide were from Dia-M (Moscow, Russia). Acetic acid and ammonium hydroxide were from Chimmed and Component-reaktiv (Moscow, Russia).

### 4.2. Bioinformatic Selection and Structural Analysis of the Therapeutic LMWLs in HSA Associated with AD

The general workflow used for selection of the clinically approved LMWLs in HSA related to AD and for their structural analysis is shown in Figure 4. The following selection criteria for the LMWLs were used: (1) ligand binding to HSA and a plasma ligand concentration not lower than plasma Aβ concentration for potential competition between the ligand and Aβ for HSA binding sites; (2) confirmed association of a ligand with AD; (3) water solubility of the ligand suited for experimental studies; and (4) efficient BBB penetration of a ligand.

The list of the substances directly related to HSA (UniProt [76] (https://www.uniprot.org, accessed on 14 March 2024) ID P02768) was extracted from DrugBank (https://go.drugbank.com/bio_entities/BE0000530, accessed on 10 June 2023), a database containing chemical, pharmacological, and pharmaceutical data on the clinically approved endogenous and exogenous substances [41,42]. The compounds with molecular mass (average mass, field “Structure”), above 900 Da (according to DrugBank definition of “small molecules”, https://dev.drugbank.com/guides/terms/small-molecule), and less than 100 Da [77] were excluded from consideration. The resulting LMWLs in HSA were filtered according to the following requirements:water solubility of the ligand (Experimental Water Solubility/Calculated Water Solubility (ALOGPS, https://vcclab.org/lab/alogps/, accessed on 10 June 2023 [78]), field “Properties” of DrugBank) should exceed 1 μM to ensure the possibility of an efficient HSA loading with the ligand; water solubility of some ligands has been estimated experimentally (Appendix A);BBB penetration of the ligand (the field “Predicted ADMET Features” of DrugBank, admetSAR, http://www.admetexp.org, accessed on 10 June 2023 [79]) should exceed 50% to ensure its efficient transfer from the bloodstream into the brain;plasma ligand concentration (manually collected from Pubmed (https://pubmed.ncbi.nlm.nih.gov, accessed on 14 March 2024) and DrugBank (taken from the field “Absorption”) should exceed 0.5 nM, which corresponds to the total plasma Aβ40 concentration [24].

The FAs were excluded from further consideration, since they had been studied in the previous work [35].

The association of the resulting LMWLs with AD was examined for each of them using the query «“Alzheimer’s disease” + “substance name”» in the Alzforum online resource (https://www.alzforum.org/papers, accessed on 10 October 2023). The substances with less than 2 references on Alzforum were excluded from the further analysis, resulting in a total number of 100 candidates.

To classify the selected LMWLs according to the expected location of their binding sites on the HSA surface, molecular docking of HSA and the ligands was performed using AutoDock Vina [44] (https://vina.scripps.edu, accessed on 14 March 2024). AutoDock Vina is a computational docking program that is based on a simple scoring function and rapid gradient-optimization conformational search. It has improved support for energy minimization (https://sourceforge.net/projects/smina/, accessed on 14 March 2024) and an efficient quasi-Newton Broyden–Fletcher–Goldfarb–Shanno (BFGS) method was used for the local optimization [44].

The crystal structure of HSA was taken from Protein Data Bank (PDB) [80] (https://www.rcsb.org, accessed on 14 March 2024): chain A of entry 1UOR. AutoDockTools software (https://autodocksuite.scripps.edu/adt/, accessed on 14 March 2024) was used for preparation of the PDB structure for the docking process, including removal of water molecules and the addition of lacking hydrogen atoms (recommended stages for preparation of molecules for docking) [81]. The three-dimensional structures of the LMWLs were taken from PubChem server (https://www.ncbi.nlm.nih.gov/pccompound, accessed on 14 March 2024) in a structure data format (filename extension .sdf) and converted to a PDB format using PyMOL (https://pymol.org/, accessed on 14 March 2024). In the case of stereoisomers, the biologically active (predominantly R) isomer was used for the modeling (for ibuprofen and for warfarin the more active (S)-isomer and both enantiomers were used). The docking model corresponding to the lowest energy of the HSA-ligand complex was chosen. The protein–ligand complexes were visualized using PyMOL v.1.6. The HSA–ligand interactions were analyzed using the protein–ligand interaction profiler PLIP [82] (https://plip-tool.biotec.tu-dresden.de/plip-web/plip/index, accessed on 14 March 2024). The numbering of the amino acid residues corresponds to PDB entry 1UOR.

Dynamic time warping algorithms implemented in the ‘dtw’ library [46] written in the R language (https://www.r-project.org/, accessed on 14 March 2024) were used to hierarchically cluster (five clusters were chosen as an optimum) the AD-associated LMWLs based on the predicted location of their binding sites on HSA molecules. The rank of a ligand in a cluster, *R*, was calculated for all clusters using the Equation (1):(1)$$R=\sum_{1}^{a}\left(\frac{f_{aa}}{n}*k\right),$$
where *f_aa_* is the frequency of occurrence of an amino acid at the binding site within the cluster, calculated as a number of ligands in the cluster whose binding site contains this amino acid; *a* is the number of amino acids in the cluster, calculated as a sum of all non-repetitive amino acids forming binding sites for all ligands in the cluster; *n* is the number of ligands in the cluster; and *k* is a coefficient equal to 0 if the amino acid is absent at the binding site, and equal to 1 if the amino acid is present in the binding site.

The aforementioned algorithms for search, collection, alignment, representation, and analysis of the data were implemented using the Python 3 (https://www.python.org/, accessed on 14 March 2024) programming language in a PyCharm v.2018 environment (https://www.jetbrains.com/pycharm/, accessed on 14 March 2024). The specialized Python libraries (Requests, BeautifulSoup) were used to form HTTP requests and parse the web pages, search, and collect the data into a local database.

### 4.3. Preparation of Recombinant Aβ

The human Aβ40 samples were pretreated prior to experimental studies essentially as described in ref. [37]. The freeze-dried Aβ40 samples were dissolved in neat TFA at a concentration of 0.5 mg/mL, followed by sonication for 30 s and TFA evaporation using an Eppendorf Concentrator plus. The dried Aβ40 samples were dissolved in DMSO at a concentration of 2 mg/mL and stored at −20 °C.

### 4.4. Solubility of HSA Ligands

The solubility of propranolol, prednisone, warfarin, levothyroxine, and mefenamic acid in a 20 mM HEPES(Tris)-HCl, 150 mM NaCl, pH 7.4 buffer was determined by the sequential addition of the ligand with constant stirring until the appearance of the precipitate. DMSO (for prednisone, levothyroxine and propranolol) or ethanol (for warfarin and mefenamic acid) was added to the solution at a concentration of up to 4–5% (*v*/*v*) to improve solubility of the ligand.

### 4.5. SPR Studies

The SPR measurements of the HSA interaction with monomeric Aβ40 were performed at 25 °C using a Bio-Rad ProteOn™ XPR36 instrument mainly according to ref. [35]. Ligand (50 µg/mL Aβ40 in 10 mM sodium acetate, pH 4.5 buffer) was immobilized on a ProteOn GLH sensor chip surface by amine coupling up to 10,000–14,000 resonance units, RUs. The remaining activated amine groups on the chip surface were blocked by 1 M ethanolamine solution. The noncovalently bound Aβ40 molecules were washed off the chip surface with a 0.5% SDS water solution until stabilization of the SPR signal. Analyte in 5 concentrations (2.5–40 µM HSA) in the running buffer (20 mM Tris-HCl, 150 mM NaCl, pH 7.4) in the presence/absence of prednisone (2.5 mM)/warfarin (1 mM)/mefenamic acid (250 µM)/levothyroxine (15 µM)/propranolol (1 mM) was passed over the chip at a rate of 30 µL/min for 300 s (association), followed by flushing the chip with the running buffer for 2400 s (dissociation). The sensor chip surface was regenerated by the passage of 0.5% SDS water solution for 100 s. The kinetic and equilibrium association/dissociation constants for the HSA–Aβ40 interaction in the absence of the ligands were determined with and without addition of 5% DMSO/ethanol.

The kinetic SPR data were corrected for baseline drift and non-specific binding, and described using a heterogeneous ligand model (Equation (2)) (Aβ40 and HSA serve as a ligand (L) and an analyte (A), respectively):
(2)$$\begin{array}{ccccccc} & k_{a1} & & & k_{a2} & & \\ L_{1}+A & \to & L_{1}A & L_{2}+A & \to & L_{2}A & \\ & \leftarrow & & & \leftarrow & & \\ & k_{d1} & & & k_{d2} & & \\ & K_{D1} & & & K_{D2} & & \end{array}$$
where *k_a_* and *k_d_* are kinetic association and dissociation constants, respectively; *K_D_* are equilibrium dissociation constants; and indexes 1 and 2 refer to the distinct binding sites on the ligand. The *k_a_*, *k_d_*, and *K_D_* values were estimated using Bio-Rad ProteOn Manager™ v.3.1 software (Bio-Rad Laboratories, Inc., Hercules, CA, USA). The estimates were performed for each analyte concentration, followed by their averaging (standard deviations are indicated). The free energy change accompanying the HSA–Aβ40 interaction (ΔG) was calculated as follows: Δ*G_i_* = −*RT* ln(55.3/*K_Di_*), *i* = 1,2.

## 5. Conclusions

Our previous works have revealed the precious feature of the specific HSA ligands, such as ibuprofen [37] and serotonin [36], to improve HSA affinity for Aβ, which may contribute to prevention of AD. Here we extended search for the substances with a similar effect to all approved drugs. The careful filtering of the DrugBank according to several rational criteria, followed by the selection of the candidates relevant to AD, gave rise to a panel of 100 top-ranked LMWLs. Although molecular docking studies enabled their classification depending on location of their binding sites on the HSA molecule, the SPR data do not reveal strong regularities regarding the ability of the ligands belonging to different clusters to affect the HSA–Aβ interaction. Thus, all 100 candidate ligands are of potential value in this respect. Among the LMWLs studied in this work, prednisone has the most pronounced favorable effect on HSA affinity for monomeric Aβ, with a 13-fold decrease in the equilibrium dissociation constant. Taking the data from this study and our previous reports together, we conclude that HSA ligands can either prevent or promote their interaction with monomeric Aβ. The introduction of such ligands into the bloodstream may affect the HSA-dependent Aβ deposition in the brain, which in turn may affect the rate of amyloid plaque accumulation.

Meanwhile, HSA buffers Aβ in the bloodstream and is able to inhibit Aβ fibrillization in the brain [24] via binding not only monomeric Aβ, but also its oligomeric forms and protofibrils [83,84,85], indicating the need to verify the ability of the candidate ligands to favor the inhibition of Aβ fibrillation by HSA. The ability of LMWLs to modulate the HSA effects on the Ab fibril formation process has been previously shown for warfarin, palmitic acid, cholesterol, and ibuprofen [37,86]. Moreover, some of the HSA ligands may also inhibit Aβ fibrillation, as shown for serotonin [87]. Thus, the search for the optimal therapeutic ligand involves extensive experimental studies and filtering of the candidates in accordance with the abovementioned requirements. However, the present work provides a basis for these studies aimed at the repurposing of the approved drugs for prevention and treatment of AD. Additional clinical trials should be performed for the ligands that prevent HSA–Aβ interaction (for instance, mefenamic acid and risperidone) to rule out the possibility of stimulation of Aβ depositions.

## Figures and Tables

**Figure 1 ijms-25-04975-f001:**
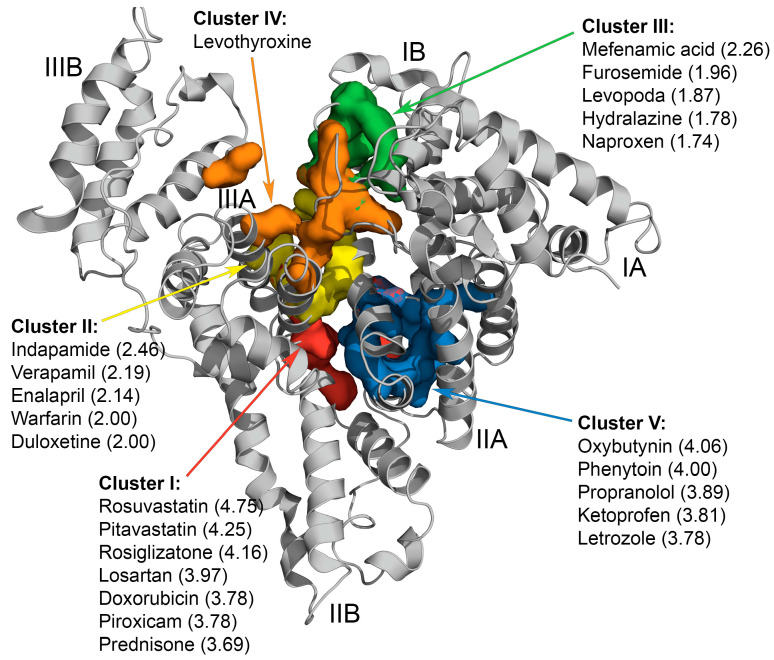
Predicted location of the key HSA residues at the binding sites (cluster I, red; cluster II, yellow; cluster III, green; cluster IV, orange; cluster V, blue) for the AD-related LMWLs clustered using the R language library ‘dtw’. The ligands of the individual clusters with highest *R* values (Equation (1)) are indicated (see Appendix A for the complete list), along with the *R* values (shown in the parentheses). The subdomains IA, IB, IIA, IIB, IIIA, and IIIB are marked.

**Figure 2 ijms-25-04975-f002:**
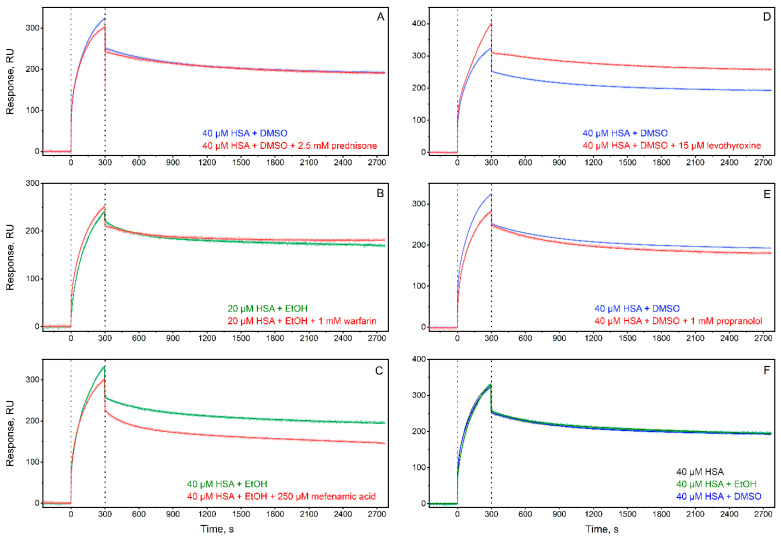
The kinetics of HSA association (0–300 s) with the Aβ40 monomer immobilized on SPR chip’s surface by amine coupling, as well as the dissociation of their complex (300–2700 s), monitored by SPR spectroscopy in the presence (panels (**A**–**E**)) or absence (**F**) of the HSA ligands shown in Table 1: 2.5 mM prednisone (panel (**A**)), 1 mM warfarin (**B**), 250 μM mefenamic acid (**C**), 15 μM levothyroxine (**D**), and 1 mM propranolol (**E**) (20 mM Tris-HCl, 150 mM NaCl, pH 7.4; 25 °C). The experimental curves are described within the heterogeneous ligand model (Equation (2)) (see Table 2 for the fitting parameters).

**Figure 3 ijms-25-04975-f003:**
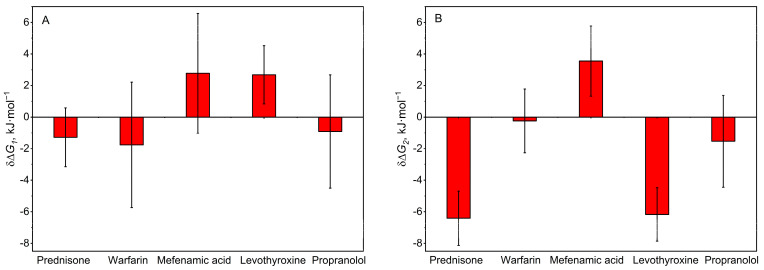
The changes in the free energy changes accompanying HSA–Aβ40 interaction (*δ*Δ*G_i_*, *i* = 1, 2) induced by addition of the specific HSA ligands (see Table 1), calculated from the SPR data shown in Table 2 (Δ*G_i_* = −*RT* ln(55.3/*K_Di_*), *i* = 1, 2). Panel (**A**) corresponds to K_D1,_ panel (**B**) K_D2_.

**Figure 4 ijms-25-04975-f004:**
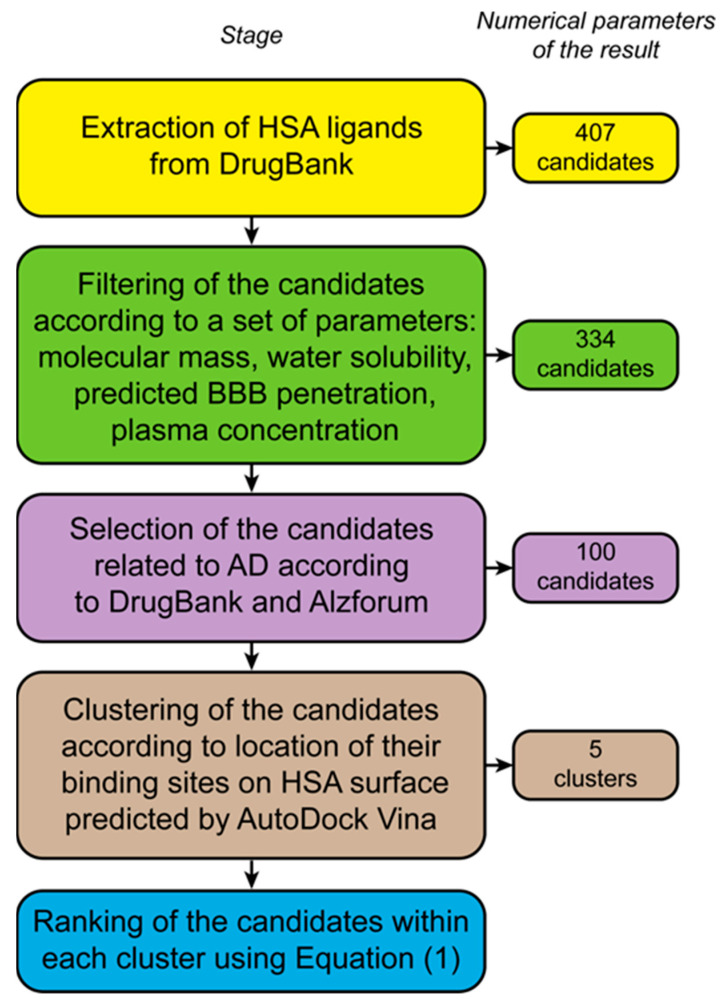
The scheme of the bioinformatic search for the AD-related therapeutic HSA ligands, their filtering, structural analysis, and ranking.

**Table 1 ijms-25-04975-t001:** Information on the AD-associated LWMLs selected for experimental examination of their influence on HSA–Aβ interaction.

Predicted Cluster on HSA Molecule	Drug	Drug Class	Application Area	Equilibrium Association Constant for the Drug–HSA Interaction	Calculated Occupancy of HSA Binding Sites for 2.5–40 µM HSA
I	Prednisone	Corticosteroid	Transplantology; treatment of allergy, inflammation, infection, cancer, endocrine, autoimmune conditions	K = 1 × 10^3^ M^−1^ [47]	71%
II	Warfarin	Antithrombotic agent	Thromboembolism treatment	K_1_ = 2 × 10^5^ M^−1^ K_2_ = 5 × 10^4^ M^−1^ [48]	site1: 99–99.5%; site2: 98%
III	Mefenamic acid	non-steroidal anti-inflammatory agent	Analgesia, treatment of inflammation и fever	K_1_ = 4 × 10^5^ M^−1^ [49] K_2_ = 1 × 10^5^ M^−1^ [50]	95–96%
IV	Levothyroxine	Thyroid hormone	Treatment thyroid diseases including hypothyroidism and cancer	K = 1 × 10^5^ M^−1^ (4 sites) [51]	9–41%
V	Propranolol	Beta blockers	Cardiology, including hypertension and myocardial infarction	K = 1 × 10^4^ M^−1^ (2 sites) [52]	90–91%

**Table 2 ijms-25-04975-t002:** Parameters of HSA-Aβ40 interaction in the presence/absence of the HSA ligands shown in Table 1 determined by SPR technique using the heterogeneous ligand model (Equation (2)).

Ligand/Additive	k_a1_ × 10^−2^, M^−1^s^−1^	k_d1_ × 10^4^, s^−1^	K_D1_ × 10^7^, M	k_a2_ × 10^−2^, M^−1^s^−1^	k_d2_ × 10^4^, s^−1^	K_D2_ × 10^6^, M
Without ligand + DMSO	2.7 ± 2.6	0.216 ± 0.007	0.64 ± 0.48	8.9 ± 9.0	26 ± 13	5.0 ± 3.4
Without ligand + ethanol	5.6 ± 4.7	0.27 ± 0.20	1.2 ± 1.6	13 ± 8	34 ± 11	3.2 ± 1.5
Prednisone + DMSO	2.73 ± 0.96	0.11 ± 0.05	0.38 ± 0.05	36 ± 8	13.6 ± 1.2	0.38 ± 0.05
Warfarin + ethanol	7.9 ± 3.7	0.50 ± 0.39	0.57 ± 0.43	24 ± 28	32 ± 11	2.9 ± 1.9
Mefenamic acid + ethanol	3.3 ± 1.6	0.97 ± 0.56	3.6 ± 2.1	3.6 ± 3.4	31 ± 12	13 ± 10
Levothyroxine + DMSO	0.575 ± 0.011	0.109 ± 0.006	1.90 ± 0.14	27.0 ± 1.6	11.20 ± 0.12	0.41 ± 0.02
Propranolol + DMSO	22 ± 31	0.204 ± 0.010	0.45 ± 0.55	25 ± 34	18.8 ± 0.2	2.7 ± 2.6

## Data Availability

Data are contained within the article and Appendix A.

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
