# Peer review of "In Search for Low-Molecular-Weight Ligands of Human Serum Albumin That Affect Its Affinity for Monomeric Amyloid β Peptide"

_ijms, 2024, doi:10.3390/ijms25094975_

Round 1

Reviewer 1 Report

Comments and Suggestions for Authors

Please consider the attached file.

Reviewer 2 Report

Comments and Suggestions for Authors

Title; In Search for the Low-Molecular-Weight Ligands of Human Serum Albumin that Affect its Affinity for Monomeric Amyloid β Peptide

Comments;  In my view, the results obtained in this study are worthy for publication. The manuscript needs major essential revision before publication. I would like to overview the revised version of the manuscript. I have the following comments/suggestions for authors to address before final decision on the manuscript.
1. Clarify the rationale behind the selection of human serum albumin as a natural depot for amyloid β peptide binding and its potential role in Alzheimer's disease prevention. Providing a comprehensive background on the interaction between Aβ and HSA would enhance the significance of the study.
2. Specify the bioinformatic search criteria used to identify clinically approved AD-related low-molecular-weight ligands of HSA. Transparency in the search methodology will ensure reproducibility and facilitate future research in this area.
3. Provide detailed information about the classification criteria for categorizing the top 100 candidate LMWLs into the five clusters. Clear definitions of these clusters and the rationale behind their classification will strengthen the study's methodology.
4. Describe the experimental validation methods used to assess the impact of candidate LMWLs on HSA affinity for Aβ. Including details about the experimental protocols, controls, and statistical analyses will enhance the rigor and reliability of the results.
5. Discuss the potential limitations of the bioinformatic approach and classification methodology used in the study. Addressing these limitations will help interpret the results accurately and guide future research directions.
6. In the Introduction section the author should refer to the research paper and comment on recent in-silico techniques. It will be good information for the readers. I would like to recommend several papers, among many others, providing further explanation on this topic:PMID: 31279631 PMID: 21989830 PMID: 23383724
7. Authors have suggested to provide additional information on the structural characteristics of the specific representatives of each cluster and their predicted binding sites on the HSA surface.
8. Authors have not discussed the clinical relevance of the observed changes in equilibrium dissociation constants for the HSA-Aβ interaction induced by different LMWLs. How these changes may impact Aβ accumulation and AD progression in vivo.
9. Authors should explore the molecular mechanisms underlying the bidirectional effects observed with certain LMWLs, such as prednisone and levothyroxine. Investigating the specific molecular interactions involved will elucidate their potential therapeutic mechanisms.
10. Authors have advised to discuss the safety profile and potential side effects of the identified LMWLs, particularly those with bidirectional effects on HSA-Aβ interaction. Assessing the safety and tolerability of these compounds is essential for their clinical translation.
11. Authors have to conclude with a summary of the key findings and their implications for AD prevention and treatment. In addition provide the significance of the study in advancing our understanding of HSA-Aβ interaction and its therapeutic potential in AD management.
12. The author should clarify the methods used for the bioinformatic search and classification of LMWLs of HSA based on their predicted binding sites.
13. The author should include more details about the criteria for selecting the top 100 candidate LMWLs and how they were ranked within the clusters.
14. The author should provide a more comprehensive explanation of the experimental validation process used to assess the impact of candidate LMWLs on HSA affinity for amyloid β peptide (Aβ).
15. The author should discuss the potential mechanisms by which LMWLs improve HSA affinity for Aβ and how this may contribute to Alzheimer's disease prevention.
16. Expand on the rationale behind choosing specific LMWLs for experimental validation and their relevance to Alzheimer's disease.
17. The author should include information about the structural characteristics of the selected LMWLs and how they may influence their binding affinity for HSA and Aβ.
18. The relevance of LMWLs in current AD treatment strategies and potential synergistic effects with existing therapies should be discussed.
19. Provide more detailed descriptions of the methods used for the expression and purification of recombinant proteins, as well as the preparation of the FA-free HSA. This will enhance the reproducibility and transparency of the experimental procedures.
20. Define the parameters used for filtering candidate ligands more clearly. For example, elaborate on how the predicted BBB penetration and water solubility values were determined, as well as the rationale behind the chosen thresholds.
21. Consider discussing the limitations of predicting binding sites solely through computational methods.
22. Address the challenges associated with the low solubility of levothyroxine in the assay buffers and its potential impact on the experimental results.
23. Offer a more comprehensive interpretation of the SPR data, particularly regarding the observed changes in HSA affinity for Aβ40 in the presence of different ligands.
24. Compare the obtained results with existing literature data on similar experiments, if available.  

Comments on the Quality of English Language

Minor editing of English language required

Reviewer 3 Report

Comments and Suggestions for Authors

In Search for the Low-Molecular-Weight Ligands of Human Serum Albumin that Affect its Affinity for Monomeric Amyloid β Peptide by Evgenia I. Deryusheva et al

Comments to the Editor-in-Chief and Authors

These authors have peviously demonstrated the ability of specific low-molecular-weight ligands (LMWLs) of HSA toimprove its affinity for Aβ. Here they develop a bioinformatic approach for the clinically approved AD-related LMWLs of HSA; however, it is not possible to extrapolate their findings to the clinical levels in patients with Alzheimer disease without real data in these patients; they selected good drugs candidates according to the predicted location of their binding sites on HSA surface, wich were demonstrated by the experimental validation of their impact on HSA affinity for Aβ. The top 100 candidate LMWLs were classified into the five clusters. The specific representatives of the different clusters exhibit dramatically different behavior, with 3- to 13-fold changes in equilibrium dissociation constants for the HSA-Aβ40 interaction: prednisone favors HSA-Aβ interaction, mefenamic acid  shows the opposite effect; interestingly, levothyroxine exhibits the bidirectional effects. They suggets that LMWLs of HSA chosen provide a basis for drug repurposing for AD prevention, and for  search for the medications promoting AD progression. However, their findings, which are interesting, can not be extrapolate to patients without real findings (again).

My mean complain is that the discussion should be more focused in Alzheimer disease and these drugs without information about other neurological diseases. The conclusion should be reduced and these smalll comments included in the R1 version.

In general, these bioinformatic tools are interesting and can predict the effect of drug and the binding to albumine, which are predictive and could be interesting from a molecular view point.

The excel for tables1 and 2 with all drugs should be included as supplementary files but not include within the mean manuscript. Please, included it as anexo.

Thus, my Decision is minnor revision

General Comments

Shall you indicate posible adverse to drugs of this indicated afirmation in line 59 ¨. Preliminary clinical trials confirm the effectiveness of AD treatment by replacement of the patient's serum albumin with its pharmacological preparation via plasmapheresi (plasma exchange (PE)) [19,23]. Is there any reported adverse effect associated to this replacement?

Please, explain the criterio followed for the identificaton of 100 compounts of DrugBank and /or Alzforum in your box diagram?

Line 116. Also, explain the reason by which these authors have excludedcompounds with molecular mass (average mass, field “Structure”), above 900 Da and less than 100 Da [42] were excluded from consideration.

Line 112. Why they indicate that should exceed 1 μM to ensure the  possibility of efficient HSA loading with the ligand. Is posible to use lower concentrations?

Line 135. I don`t understand the reason of The substances with less than 2 references on Alzforum were excluded from the further analysis, resultiing in a total number of the candidates of 100. Is not enough 1 reference? Have you consult other websides for this information?

Line 144. Please, also explain for a general audience the exclusion of for the docking process, including removal of water molecules and addition of lacking hydrogen atoms.

Line 175. Indicate also  the Preparation of recombinant Aβ The human Aβ40 samples were pretreated prior to experimental studies essentially as described in ref. [32] but with more details for better undestanding of a general audience.

Line 217. Although DrugBank database [40,41] was used as a source of the clinically approved HSA ligands related to AD, this feature does not mean that could really occur in patients with different degrees of Alzheimer disease at the clinical levels.

The BBB penetration is not real with this model since astrocyes, perycutes and astrocyes interacions are cue factors for the study of BBA penetrance by drugs.

Line 228. In addition, they also also used drugs as as atidiabetic drugs (rosiglitazone), non-steroidal anti-inflammatory drugs (ibuprofen,9 meloxicam), neuroleptics (risperidone), vitamins (vitamin A, thiamine), antibiotics (tetracycline, ampicillin), and hormones (testosterone, estradiol. However, clinical findings reported the risk of dementia in patients with diabetes is 1 %, which mean atidiabetic drugs (rosiglitazone) análisis should be removed in this predictive study.

Also, explain the Auto 235 Dock Vina [43], reflected in this cyte 43 for a general audience (IJMS readers). The predictive sites of binding to different drugs has been already published before.

Line 272. Explain how these differentd rug conentrations used for the SPR studies in table 1 cudl affect the clinical efficaty ot this drugs. This table-1 shows Information on the AD-associated LWMLs selected for experimental examination of their influence on HSA-Aβ interaction.

Line 317. The preclinical findings are interesting but these study does not infere any clincal translation of their findings. This study is focus on Alzheimer disease. Thus, remove alusive information in terms of Parkinson disease (PD). Remove this parragrah in the discussion ¨ For example, tetracyclines and some polyphenols are  able to interfere with aggregation of several unrelated amyloidogenic proteins, such as  α-synuclein (associated with Parkinson’s disease), amyloid polypeptide (type-2 diabe- 318 tes), and transthyretin (senile systemic amyloidosis, familial amyloid polyneuropathy 319 and cardiomyopathy) [57–60].

Line 346. They also indicated ¨ This suggestion is supported by the data presented in Table 3: several members of the different clusters exert different effects on HSA affinity to Aβ. For instance, mefenamic acid (cluster III) favors dissociation of HSA-Aβ40 complex, while prednisone (cluster I) promotes this interaction¨. Please, explain from a molecular view point these discrepances between these drugs.

I have not found evidences for clinical improvements by mefenamic acid in Alzheimer disease patients. We also indicated in table-3 (Table 3. Summary of the effect of the AD-related LMWLs of HSA on its interaction with Aβ and 364 data on the role of these ligands in AD progression).

Line 384- The ability of LMWLs to modulate the HSA effects on the Ab fibril formation process is not new since it has been previously reported for warfarin and ibuprofen [32,79]

These authors also indicated that risperidone could prevent AD in patients with psychosis. However, it is possible to distinguish psychosis or Alzheimer disease progression in this cyte study.

This study is interesting but these authors should remove either clinical extralopation without data in patients. In fact, all data are speculative from matematical models without data from rodent models of Alzheimer disease or findings in treated patients with these drugs. Their alusion in clinical findings in tables are fine but it not possible to extrapolate these bioinformation data to the clnical data without real findings in Alzheimer disease patients.

The discussion must be improveed following these comments as well as, please, reduce the conclusion with your relevant findings in your study.

Thanks¡

Comments on the Quality of English Language

Please, revise your english style.

Round 2

Reviewer 2 Report

Comments and Suggestions for Authors

The draft is not improved based on the comments, I can not recommend the revised daft for the next stage.

Comments on the Quality of English Language

Moderate editing of English language required